# The Role of Extracellular Matrix in Human Neurodegenerative Diseases

**DOI:** 10.3390/ijms231911085

**Published:** 2022-09-21

**Authors:** Panka Pintér, Alán Alpár

**Affiliations:** 1Department of Anatomy, Semmelweis University, 1113 Budapest, Hungary; 2SE NAP Research Group of Experimental Neuroanatomy and Developmental Biology, Hungarian Academy of Sciences, 1051 Budapest, Hungary

**Keywords:** sclerosis multiplex, Alzheimer’s disease, Parkinson’s disease, neurodegenerative disorder

## Abstract

The dense neuropil of the central nervous system leaves only limited space for extracellular substances free. The advent of immunohistochemistry, soon followed by advanced diagnostic tools, enabled us to explore the biochemical heterogeneity and compartmentalization of the brain extracellular matrix in exploratory and clinical research alike. The composition of the extracellular matrix is critical to shape neuronal function; changes in its assembly trigger or reflect brain/spinal cord malfunction. In this study, we focus on extracellular matrix changes in neurodegenerative disorders. We summarize its phenotypic appearance and biochemical characteristics, as well as the major enzymes which regulate and remodel matrix establishment in disease. The specifically built basement membrane of the central nervous system, perineuronal nets and perisynaptic axonal coats can protect neurons from toxic agents, and biochemical analysis revealed how the individual glycosaminoglycan and proteoglycan components interact with these molecules. Depending on the site, type and progress of the disease, select matrix components can either proactively trigger the formation of disease-specific harmful products, or reactively accumulate, likely to reduce tissue breakdown and neuronal loss. We review the diagnostic use and the increasing importance of medical screening of extracellular matrix components, especially enzymes, which informs us about disease status and, better yet, allows us to forecast illness.

## 1. Introduction

The expression “extracellular matrix” refers to a network of proteins and complex sugar molecules built up around the cellular components of every tissue. Throughout the past decades, our concept of the role of the extracellular matrix has dramatically changed. It became clear that the extracellular matrix dynamically shapes parenchymal function instead of passively supporting the cells. The extracellular matrix tightly regulates the micromilieu around cells and influences signal transduction, thereby contributing to virtually every aspect of cell life, both in physiological and pathological circumstances, such as development, migration, inflammation, healing processes, fibrosis or tumorigenesis.

We soon recognized that—similarly to any other organs—the extracellular matrix is an integral part of the brain nervous tissue as well, which vastly improved our understanding about synaptic structure and function, the locus of inter-neuronal connections. Whilst neurons and surrounding glial cells are mandatory cellular components of the synapse, previously referred to as the tripartite synapse paradigm [1], it has become increasingly evident that non-cellular perisynaptic components critically contribute to synaptic development and function, which sets up the model of the tetrapartite synapse [2,3,4]. Indeed, brain extracellular matrix establishes morphologically and biochemically specialized forms of extracellular matrix around the somatodendritic or synaptic compartments of neurons, called perineuronal nets or axonal/perisynaptic coats, respectively [5,6,7,8]. Of note, extracellular matrix components also appear in the neurovascular unit and the immune system of the central nervous system, which shape neuronal function from further perspectives [9,10]. In sum, the versatility of matrix components in diverse perineuronal domains unveiled new perspectives to understand brain chemical ultrastructure and morphology, which also turned our attention towards their role in health and disease.

This review provides insight into the change of the brain extracellular matrix in disease. We first give a short overview about the major components, location/compartmentalization and phenotypic appearance of this specific extracellular substance and subsequently spotlight their role and typical changes in disease. We address the intriguing question of whether reassembly of the extracellular matrix is a cause or consequence of neuronal degeneration and subsequent malfunction in major neurodegenerative diseases, identify their triggers and present the diagnostic and therapeutic use of monitorable extracellular matrix-related changes in patients.

## 2. Structure, Composition and Metabolism of the Extracellular Matrix in the Central Nervous System

When classifying the brain extracellular matrix, two major viewpoints emerge. Phenotypical characterization goes back to Camillo Golgi and Ramón y Cajal, who were first to recognize perineuronal nets over 100 years ago [11]. Further, specifically shaped and oriented matrix domains were identified around distinct neuronal compartments, such as perisynaptically or around the nodes of Ranvier. On the other hand, the advent of high-resolution imaging of tiny pericellular domains visualized by their immunochemical make-up allowed us to identify not only shape and location, but also the biochemical compounds of these structures. Based on previous extensive descriptions [12,13,14], we first give a short overview about how matrix composites blueprint brain perineuronal matrix phenotype.

In general, extracellular matrix is built up by fibrous proteins (such as elastins or collagen) embedded in an amorphous gel formed by nonfibrous components [15,16]. These fibrous components form a highly organized scaffold that connects to the cells’ surface by adhesive molecules, called extracellular matrix receptors. In contrast to most organs, extracellular matrix of the central nervous system is best characterized by the predominance of nonfibrous components [17,18]. Most of these nonfibrous components, typically glycoproteins, are formed by a core protein (typically aggrecan, brevican and versican), to which carbohydrate side chains attach. Based on their biochemical qualities (i.e., whether N- or O-glycosylation takes places during synthesis or a different domain sequence [19,20]) and on the protein/carbohydrate ratio of the carbohydrate side chains, glycoproteins fall into subcategories. Proteoglycans form a major subcategory, characterized by high carbohydrate content and amino sugar-containing polysaccharide side chains [21,22,23]. These amino sugar-containing polysaccharide side chains are commonly referred to as glucosaminoglycans (GAGs), which are long unbranched molecules with repeating disaccharide units composed of uronic acid and an amino sugar. Hyaluronic acid—also called hyaluronan, often referred to as the “simplest” GAG because of its lack of sulfation—is a large polymer (100–10,000 kD) formed by repeats of β-1,3-N-acetyl-glucosamine and β-1, 4-glucuronic acid disaccharide units. Due to its interaction with proteins [24,25,26,27] which link core proteins to this very long GAG (like hyaluronan and proteoglycan link protein (HAPLN1) links aggrecan to hyaluronan), we consider hyaluronic acid as the backbone of extracellular matrix. This stands also for the nervous tissue, where chondroitin sulfate proteoglycans represent the group of sulfated proteoglycans which attach to hyaluronic acid through link proteins. Basic components of the non-fibrous “ground-substance” of the extracellular matrix in the central nervous system fall into distinct categories. Typical substances [28,29,30,31] are summarized in Figure 1. 

Schematic representations of molecules were redrawn based on previous publications [32,33,34,35,36,37,38,39,40,41,42,43,44,45].

Most prominent proteoglycan representatives belong to the lectican family, which include aggrecan, versican, neurocan and brevican [13], the last two of which occur only in the central nervous system.

Brain extracellular matrix is not permanent, its structure and composition undergoes major changes during pre- and postnatal development [8,46,47], which allows neurons to adapt in health and disease. Matrix remodeling is achieved by the tight regulation of production and degradation of its components; whilst this metabolism is normally slow, select challenges like growth [48], cell division [49] and migration [50,51], inflammation [52,53] or injury [54,55,56] enhance matrix transformation. Many enzymes contribute to this process, most notably members of the matrix metalloprotease family (MMPs) and their regulators: tissue inhibitors of metalloproteinases (TIMPs), tissue plasminogen activator (tPA), plasminogen activator inhibitor, ADAMTS family, the hyaluronidase/chondroitinase family [57,58] and their various subtypes [59,60,61,62,63,64,65,66]. MMPs are endopeptidases which have a dominant role in extracellular matrix degradation in vivo. Irrespective of subtypes, MMPs have zinc content of their active site which is the principal target of their specific inhibitors; TIMPs bind here to chelate their zinc content [67,68] without subtype selectivity [69,70,71]. Of note, TIMP-3 emerges as a more potent inhibitor of some of the ADAMTS proteins [72,73], another subtype of metalloproteases, which typically catalyze the degradation of proteoglycans belonging to the lectican family [59]. The serine protease tPA—besides its role in hemostasis—catalyzes the plasminogen-plasmin conversion in the central nervous system, thereby shaping laminin degradation and activation of trophic factors such as chemoattractant protein-1, nerve growth factor (NGF) and brain-derived neurotrophic factor (BDNF). Moreover, tPA itself interacts with NMDA receptors to influence signal transduction, microglia activity [74] and blood–brain barrier permeability [75,76,77]. In addition to their inhibitory and activating enzymes, enzymes involved in extracellular matrix degradation are tightly regulated by various other mechanisms, such as: *(i)* degraded matrix proteins activate previously bound tissue growth factors [78] that stimulate the restoration of ECM and tissue growth; *(ii)* secreted inactive precursors can be later activated by limited proteolysis (like proMMP to MMP and plasminogen to plasmin; or *(iii)* the local limiting of proteinase effect by binding them to membrane receptors (eg. MMPs) [79,80,81].

Compared to MMPs, enzymes showing chondroitinase activity have minor significance in vivo, although they are present in humans and other higher organisms and contribute to extracellular matrix turnover [82,83]. Of note, enzymes of the chondroitinase family are widely used both in in vitro experiments and in animal studies to demonstrate the effect of elimination of extracellular matix components. One of the most frequently used matrix demolishing enzymes is chondroitinase ABC, a bacterial protein, which acts by degrading the glycosaminoglycan side chains of chondroitin sulfate proteoglycans (CSPGs) [84,85]. Importantly, chondroitinases emerge as promising therapeutic agents in various pathological conditions of the central nervous system, such as formation of scar tissue in the spinal cord [86,87,88,89]. 

## 3. The Phenotypic Appearance of the Brain Extracellular Matrix

The architectural, physical and functional differences of mesenchymal tissues rely on the vast diversity of components which accumulate in the abundant extracellular space. The advent of electronmicroscopical analysis of the nervous tissue suggested that the dense neuropil—i.e., the network of neurites, dendritic and glial cell processes and the microvasculature—leaves hardly any space for extracellular components free [11]. Nevertheless, in addition to the perivascular basement membrane—a critical component of the blood–brain barrier [19,90,91]—condensed forms of extracellular matrix assemblies were identified around select neuronal compartments based on their phenotypical appearance and chemical assembly.

As a first step, perineuronal nets were described by Camillo Golgi, but fell into oblivion for decades after Golgi lost his famous debate with Camón y Rajal over the basic concept of the nervous system and the cellular theory became widely accepted [11]. Rediscovery of perineuronal nets happened during the 1960s when a PAS^+^ material was described around the soma and dendrites of some neurons [92]. Of note, long before Golgi and Cajal’s time, a similar debate took place. For centuries, scientists had believed that the basis of life were “fibers” (later described as “connective tissue”), and life spontaneously arose from those fibers under certain conditions. During the first decades of the 19th century, it became evident that life could arise only from something already alive and that cells contributed largely to tissue structure. Eventually, Virchow stated in the late 1850s that cells only arise from cells, and non-cellular components of tissues are the products of cells [92]. Our view on the central nervous system—and actually on the whole body today—is based on Cajal’s and Virchow’s revolutionary ideas, including disease development.

Perineuronal nets are indeed distinct phenotypic matrix assemblies which enwrap the somatodendritic compartments of neurons, depending on neuronal type and brain area. Thus, perineuronal nets typically surround parvalbumin-containing, fast-firing interneurons [47,93,94]). Their distribution throughout the cortex, for example, varies: they occur in the neo- but not in the entorhinal cortex [95,96]); furthermore, their density in the different neocortical areas is typical and differs [95]. Irrespective of location and surrounded cell type, the backbone of perineuronal nets is aggrecan [6,97], associated with hyaluronic acid and tenascins [7,98,99,100,101]. A row of classification studies investigated and attributed diverse functions to the presence of perineuronal nets which, notably, forecasted their role in disease development. Thus, maintenance of the ideal micromilieu [101,102,103], which protects neurons against stress [104], against degeneration [101] and even against mechanical shock [105]. The role of perineuronal nets in synaptogenesis, control of plasticity or stabilizing long-term memory was also addressed.

In addition to perineuronal nets which surround the bulk neuronal compartment, tiny, isolated ring-like structures were identified later around synapses which were not embedded in perineuronal nets. These structures, termed axonal coats [99,106], ensheath preterminal fibers and synaptic boutons [107] to ensure a local scaffold for individual synapses, and are based mainly on brevican in contrast to the aggrecan-based perineuronal net [106,108]. Finally, typical extracellular matrix assemblies appear around the axon initial segment and at the nodes of Ranvier [47,109,110]. These structures ensure a chemically specified microenvironment at non-synaptic neuronal loci, to help conduction [109]. 

We may have the misleading impression that the exploration of extracellular matrix phenotype in the central nervous system has reached its zenith in the last decade. Albeit, while the major chemical and morphological characteristics of perineuronal matrix assemblies are known, several intriguing questions emerge including, but not limited to: *(i)* if functional change of the synapse triggers transitional, short-term matrix re-assembly and its significance; *(ii)* if perineuronal/perisynaptic matrix assemblies go undetected presently due its unknown composition or limited amount/thickness; *(iii)* left–right brain asymmetries remained largely unaddressed; *(iv)* whilst major immunohistochemical and morphological characteristics of brain perineuronal matrix are actually identical in human and nonhuman vertebrates, does any specific composition or difference in relative content appear in the human brain?

## 4. Extracellular Matrix Components and Neurodegenerative Diseases

Shortly after the first description of the perineuronal nets by Golgi, their possible contribution to neurodegenerative diseases was suggested by Nissl and Alzheimer, and later by Besta, Belloni and Donaggio. Besta argued that perineuronal nets were distinct anatomical structures, not part of the neuron itself, and likely produced by glia [11]. Belloni and Donaggio examined various cases of dementia, diffuse gliosis and psychiatric cases in the 1930s and reported changes both in perineuronal nets and the neuropil (“diffuse net”) structure [92].

Here we attempt to describe extracellular matrix-related brain pathology from two perspectives: based on their phenotype and biochemical assembly.

### 4.1. Phenotypic Perspective

#### 4.1.1. Perineuronal Nets and Neurodegeneration

During the past years, a series of observations led us to hypothesize that perineuronal nets protect cells from degeneration [101,111]. Among human cortical areas, the motor cortices (Brodmann 4, 6) and the primary auditory cortex (Brodmann 41) contain the highest number of perineuronal nets. Areas 21, 22, 38, prefrontal areas (Brodman 10, 11, 12) and cingulate areas (Brodman 23, 24) contain less, whilst the entorhinal cortex (Brodman 28) contains no perineuronal nets. Hyperphosphorylated tau—a characteristic hallmark of Alzheimer’s disease—did not accumulate in areas abundant in perineuronal nets [101]. When examining individual cells, it became evident that neurons—pyramidal cells and interneurons alike—surrounded by perineuronal nets remained relatively intact, even in brain with robust neurofibrillary tangle deposition. Actually, perineuronal matrix restricts distribution and internalization of aggregated tau protein, at least in organotypic slice cultures [112]. The low vulnerability of neurons enwrapped by perineuronal nets is a general phenomenon: subcortical brain regions in human subjects equally show complementary distribution of neurofibrillary tangles and perineuronal nets, and subcortical cells with aggrecan-containing pericellular matrix remain devoid of tau pathology [113]. Chondroitin sulfate proteoglycans also protect against soluble β-amyloid: cortical neurons with perineuronal nets resist amyloid protein neurotoxicity in dissociated cultures [103,113,114]. This evokes interneuron-specific changes in the brain circuit: parvalbumin-containing interneurons selectively resisted β-amyloid-induced cell damage in an Alzheimer’s disease transgenic mouse model [115]. The proposed mechanism behind perineuronal nets’ ability to protect neurons is their capacity to bind metal ions which react with free radicals [104] to trigger and maintain oxidative stress. Its degradation, as that of increased enzymatic (MMP-9) activity, can therefore exacerbate cell loss in Alzheimer’s disease [116].

The role of perineuronal nets in disease development was implicated in other clinical aspects, too. The protective role of perineuronal nets was identified and a similar mechanism and cell-type specificity suggested in Parkinson’s disease, based on rodent models [117,118,119]. Further, transcriptomics insight into spinal muscular atrophy, a recessive genetic disorder, suggests that triggering molecules of perineuronal nets, including HAPLN1, are downregulated, which likely affects ventral horn motoneuron function in the spinal cord [120].

#### 4.1.2. Axonal Coats and Neurodegeneration

Symptoms of neurodegenerative diseases are typically associated with synaptic loss [121,122]. Whilst perineuronal nets offer protection for synapses from the postsynaptic side, the role of presynaptic neurons in establishing single perisynaptic matrix assemblies [123] remained largely unaddressed. Synaptic loss in Alzheimer’s disease, however, was soon associated to loss of brevican activity, including in humans [106], with brevican being the main component of axonal coats. The selective loss of brevican in these individual perisynaptic matrices of presynaptic origin was soon reconfirmed in the human hippocampus and in relevant transgenic mouse models [100]. Ever since, extracellular matrix reassembly in neurodegenerative disorders around individual synapses and driven from the presynaptic side has received little attention. However, the identification of new perisynaptic matrix components, such as chondroitin sulfate proteoglycan-5 (CSPG-5, also referred to as neuroglycan C), of presynaptic neuronal, but not glial, origin, has given new momentum to the desire to gain insight into synapse pathology [8]. Of note, microglia, a hitherto uninvestigated cellular player of the field, has emerged recently to sculpt extracellular matrix to modify the synapses within, in both health and disease [124,125]. These findings fuel the hypothesis that neurons shelter their synapses by establishing selective matrix domains around them, and a failure in this function can lead to neurodegenerative disorders [100].

#### 4.1.3. Basement Membrane and Neurodegeneration

The nervous tissue benefits from a tight vascular barrier to protect its neurons which are most sensitive to agents that are relatively harmless to most other cell/tissue types. This blood–brain barrier consists of endothelium, pericytes, astrocytic endfeet, neuronal processes, microglia and the basement membrane, which is a specialized form of the extracellular matrix [126,127]. In contrast to peripheral basement membranes, the basal lamina of the central nervous system includes an additional, astrocyte-derived component [128], which empowers the membrane with distinct functions [129]. Blood–brain barrier failure/leakage is a diagnostic sign in neurodegenerative diseases, such as Parkinson’s disease, which can be visualized by positron emission tomography (PET) [130].

It is unclear if changes in major extracellular matrix components of the basement membrane-like collagen IV, laminin, nidogen, or the heparin sulfate proteoglycans agrin and perlecan-are cause or consequence of neurodegeneration. Actually, basal lamina thickens also during normal aging, which is caused by altered expression levels of the different matrix components [129]. In neurodegenerative diseases, however, select pathomechanisms in matrix metabolism were identified which trigger/aggravate tissue damage. The evoked vascular damage leads to hypoperfusion which compromises the clearance of neurotoxins, including β-amyloid in rodent models of Alzheimer’s disease [131,132]. Electronmicroscopic analysis showed that basement lamina thickening occurs in its parenchymal (glial) but less so in the endothelial domain in human brain tissue samples affected by AD [133], which precedes vascular amyloid deposition [134] and is due to the failure of receptor-mediated transvascular clearance [135] as both proven in corresponding transgenic mouse model. Similarly, human truncated tau increases vascular permeability through activating glial cells [136]; actually, taupathy can cause blood–brain barrier damage without amyloid pathology [137]. At the molecular level, enhanced collagen IV expression was shown to bind amyloid precursor protein [138] and to prevent β-amyloid aggregation, at least in vitro [139]. Similarly, select laminin isoforms act protectively when inhibiting β-amyloid fibril formation in vitro [140]. In contrast, heparin sulfate proteoglycans accelerate β-amyloid fibrillation through binding β-amyloid on their heparin sulfate chains in vitro [141] and inhibit β-amyloid degradation [142]. Furthermore, β-amyloid inhibits heparanase in vitro, leading to a vicious cycle in the aggravation of tissue damage [143]. We know considerably less about mechanisms which evoke basal lamina changes in Parkinson’s disease: the thickened and collapsed basal membrane is probably due to defunct α-synuclein transport/clearance, which attenuates tissue pathology [129]. Similarly, the thickening of the basal lamina is an early sign in amyotrophic lateral sclerosis (observed in a transgenic mouse model of the disease, likely serving as a defense mechanism [144]).

A further aspect of blood–brain barrier disruption is that it allows inflammatory cells to enter the brain tissue [30]. Typically, both human and animal model studies show that immune cell trafficking across the blood–brain barrier is enhanced in multiple sclerosis [145,146]. Several mechanisms were identified at the molecular level: expression of the heparan sulfate-degrading enzyme heparanase is induced in infiltrating CD4^+^ T cells [147], and accumulation of versican and the 4-sulfated glycosaminoglycan side chains of chondroitin sulfate proteoglycans in the perivascular cuff boost the activity and migration of leucocytes across the blood–brain barrier, which worsens the clinical severity of multiple sclerosis [148]. 

In addition to the challenge posed by perineuronal and especially perisynaptic matrix assemblies that are presently undetectable immunohistochemically—due to their low amount or unknown composition—which could significantly improve our knowledge on disease detection and mechanism, several questions have remained largely unaddressed until now. Neuron-glia and neuron-microglia talk can be a critical trigger in matrix re-assembly, and the identification of specific inter- and intracellular downstream mechanisms can offer druggable targets in the future. It has remained unknown if the pre- or the postsynaptic neuron “cares” for its own safety and survival in neurodegenerative diseases, and whether it reflects or results in extracellular matrix rearrangements. Finally, despite our accumulating knowledge, we still know little about the dynamic composition change of perineuronal nets in disease, especially around the synapse.

### 4.2. At the Biochemical Level: Component Changes behind the Phenotypic Signs during Neurodegeneration

#### 4.2.1. Hyaluronic Acid 

Hyaluronic acid concentration typically increases in neurodegenerative diseases. In multiple sclerosis, two different types of hyaluronic acid accumulate in demyelinated lesions. Acute lesions are characterized by a smaller molecular weight form, which is thought to further elicit inflammation. On the other hand, chronic lesions are enriched in high molecular weight hyaluronan in both human and mouse tissues. This latter form plays a role in the inhibition of oligodendrocyte precursor cell maturation which ultimately leads to impaired remyelinization [149]. The focality of white matter damage is also reflected in the corresponding focal expression pattern of KIAA1199, a protein which degrades hyaluronan and is released by activated astrocytes, at least in mice [150]. Further, hyaluronan accumulation at chronic inflammation foci creates a permissive environment for autoimmunity, characterized by CD44-mediated inhibition of regulatory T-cell expansion [151,152,153]. As such, increased hyaluronic acid level in the cerebrospinal fluid of human patients emerges as a useful diagnostic marker in multiple sclerosis [154].

In Alzheimer’s disease, hyaluronic acid accumulates at the perimeter, but less so towards the center of plaques. Excess hyaluronan might be produced by reactive CD44^+^ astrocytes [153,155], which are in turn activated via their hyaluronic acid receptors [156,157], thereby creating a positive feedback loop. This vicious cycle, which leads to hyaluronan overload, affects both the perivascular and the perineuronal domains: *(i)* it disrupts the blood–brain barrier, decreases blood flow with consequent loss of glucose and oxygen-supply, but *(ii)* also affects the composition of perineuronal nets, which become less effective in terms of protecting neurons from ischemia. Abnormal production can also be the consequence of altered hyaluronan synthase pattern: hyperphosphorylated tau remodels the activity and compartmentalization of different hyaluronic acid synthase types, which finally leads to defunct perineuronal net formation [158]. Of note, the production of abnormally short hyaluronan molecules and hyaluronan degradation products trigger inflammation which mimics the mechanism described in active plaques in multiple sclerosis, and exacerbates tissue damage in the human brain affected by Alzheimer’s disease [159,160]. Enrichment of higher molecular weight hyaluronic acid in β-amyloid plaques and in chronic MS lesions, however, reflects consolidation: they bind anti-inflammatory factors (as in TSG-6 mice) to moderate inflammatory response and indicate the formation of glial scars [161,162]. 

#### 4.2.2. Proteoglycans and Neurodegeneration

The expression of proteoglycans typically increases in neurodegenerative diseases. Despite previous progress, why these essential extracellular matrix components undergo remodeling in brain pathology remains ambiguous. Instead of being byproducts during the development or restorative processes of neurodegenerative diseases, they likely accumulate to either protect/maintain function or, conversely, to trigger tissue breakdown. In fact, separating protective and causative actions of proteoglycans brain pathology is challenging: different proteoglycans act at different loci and at different stages of disease development, which can reflect an independent, complementary, synergistic or antagonistic role/effect. Much of the future research in the field must aim to understand the functional importance of these time- and locus-specific changes.

In multiple sclerosis, lecticans, (namely aggrecan, versican and neurocan) and dermatan sulfate-proteoglycans (DSPGs) are upregulated around the edges of acute lesions alongside astrogliosis. This pathological, likely reactive change is responsible for inhibition of axonal growth and remyelination. Towards the center of the lesion, lecticans and DSPGs were found to be phagocytosed by macrophages together with damaged myelin components. Increased CSPG concentration directly activates inflammatory cells and enhances their ability to migrate into the brain parenchyma in active lesions. In mice, a specific mechanism was unveiled when versican was shown to promote T helper 17 cytotoxic inflammation and to impede oligodendrocyte precursor cell remyelination [163]. Thus, reactive astrogliosis likely leads to depletion of CSPGs and other proteoglycans, which in turn contribute to neuroinflammation. Proteoglycan binding agents, such as surfen [164], fluorinated glucosamines [165] or heparin antagonists [166], can therefore effectively reduce inflammation and inhibit remyelination. Furthermore, ligands, such as the endocannabinoid 2-arachidonoylglycerol, which enhance glia differentiation, can neutralize CSPG accumulation, which offers a druggable target in multiple sclerosis therapy as seen in human and rat astrocyte culture studies [167].

A series of studies investigated proteoglycans in Alzheimer’s disease. Similarly to spinal cord lesions [168], CSPG accumulation might physically contribute to neurite degeneration and also has a negative impact on cell adhesion and growth factor distribution. CSPGs are present in neurofibrillary tangles and senile plaques with different sulfation states in both humans and rats, and quantitative analysis of these can forecast onset and progression of the disease [169]. In combination with the inhibitory receptor protein tyrosine phosphatase σ (PTPσ), CSPGs inhibit neuronal plasticity [170], which can be reversed by the application of chondroitinase ABC (ChABC) [171]. Although CSPGs restrict plasticity by forming perineuronal nets [172,173], these select matrix assemblies protect neurons from degeneration, which makes CSPGs likely protective with this specific location in Alzheimer’s disease. Keratan sulfate proteoglycans (KSPGs), primarily located at synapses and dystrophic neurites within neuritic plaques [142], lack high sulfation in Alzheimer’s disease, which weakens interneuronal communication and impairs learning and memory in patients [174].

High and low molecular weight heparin promote or inhibit β-amyloid aggregation/structure stabilization, respectively [175]. HSPGs also accumulate in the brain of Alzheimer’s disease patients, with agrin effectively enhancing β-amyloid fibril formation and inhibiting its proteolysis via its nine protease-inhibiting regions [121,176]. Sulfation-specificity emerges as a regulatory factor: highly N-sulfated and O-sulfated HSPGs can trigger fibril initiation [177] and promote stabilization [178]. Similarly to CSPGs, HSPGs not only appear in the close neighborhood of β-amyloid plaques, but also co-localize with hyperphosphorilated tau. Actually, HSPGs regulate cellular immune responses to tau protein monomers, which creates a neuroinflammatory environment for tangle formation [179]. The early accumulation of HSPGs, as well as the related molecular mechanisms in disease development, made HSPGs primary and key players in the unifying hypothesis of Alzheimer’s disease [180,181]. On the other hand, there is supporting evidence that HSPGs take part in the pathway responsible for β-amyloid uptake, likely as a clearance mechanism.

Agrin appears as critical proteoglycan in Parkinson’s disease as well: it enhances the insolubility of alpha-synuclein (at least in vitro) and co-localizes with it in Lewy bodies in the human tissue [182]. Both Lewy body and amyloid plaque formation is accelerated in the presence of a “central core” with high concentrations of HSPGs [183,184]. 

Glucosaminoglycans (side chains of proteoglycans) are especially critical molecules in alpha-synuclein formation; their high concentration, long polymer isoforms, high protein:glycosaminoglycan ratio, charge density and select orientation of sulfate groups trigger and promote pathology. Different types of glucosaminoglycans lead to changes in fibril structure and cytotoxicity and influence the cellular uptake of alpha-synuclein [183]. However, glucosaminoglycans’ action is not limited to promoting deposition. Depending upon their sulfation status—which is compromised in Alzheimer’s disease—they can bind and potentiate activities of growth factors including FGF-2, VEGF and BDNF in human hippocampus samples [185].

#### 4.2.3. Glycoproteins and Neurodegeneration

The glycoprotein composition of the extracellular matrix reflects both phase/status and the type of neurodegeneration. Expression of different glycoproteins may change oppositely, likely reflecting their different roles or fate in the disease mechanism. In multiple sclerosis, tenascin concentrations decrease in the active lesions, while fibronectin and vitronectin are upregulated [30]. Loss of tenascins in active lesions is indicative of enzymatic breakdown of the matrix, whereas increased production likely reflects a protective mechanism. Excess glycoprotein production, however, can lead to disorganized matrix depletion and inhibition of restorative processes like remyelination in multiple sclerosis [186]: pathological fibronectin aggregates impair remyelination by inducing pro-inflammatory features in macrophages [187]. Compounds, such as gallic and vanillic acid, reduce tenascin expression and block neuroinflammation [188]. Whilst active demyelinating lesions drive glycoprotein shift, chronic lesions show near-normal or moderately increased glycoprotein levels, due to reactive matrix production of astrocytes.

Glycoprotein shifts are also typical in Alzheimer’s disease. In corresponding animal models, tenascin-C deficiency reduces pro- but enhances anti-inflammatory activation, associated with a reduced cerebral β-amyloid load and higher levels of the postsynaptic density protein 95 (PSD-95) [189]. Tenascin-C is likely involved in β-amyloid plaque pathogenesis, since it associates with cored plaques, but not with diffuse β-amyloid plaques lacking amyloid cores, reactive glia or dystrophic neurites [190]. At the same time, tenascin-R emerged as a critical component of perineuronal nets, because it forms cross-links between CSPGs. In this locus and bondage, tenascin-R restricts both distribution and internalization of tau; hence, acts protectively in mice organotypic slice cultures [112]. Of note, chondroitin sulfate oligosaccharides themselves—as gained by recombinant chondroitin sulfate endolysis—suppress β-amyloid oxidative stress and relevant injury, at least in vitro [191]. Glycoproteins appear in the cerebrospinal fluid, which raised the possibility of using them as markers in disease progress. Nevertheless, only tenascin-C and tenascin-R concentrations were significantly higher in women, but not in men, in Alzheimer’s-diseased individuals, based on the Aβ42/40 cutoff [192]. 

Amongst other glycoproteins, thrombospondin-1 attracted attention because it supports synapse maintenance [193]. Thrombospondin-1 decreases in humans and animal models of Alzheimer’s disease [194] in an autophagy-dependent manner, and β-amyloid application decreases thrombospondin-1 secretion [195]. Consistently, application of thrombospondin-1 rescues dendritic spine and synapse loss and mitigates the β-amyloid-mediated reduction of synaptic proteins and related signaling pathways. This rescue effect has been exploited when paracrine secretion of thrombospondin-1 from human umbilical cord blood-derived mesenchymal stem cells was used to attenuate β-amyloid induced synaptic dysfunction [196]. Thrombospondin-1 also acts through an alternative mechanism: it protects against β-amyloid-induced mitochondrial fragmentation, thereby preventing cellular dysfunction [197]. Neurexin 1b, a neuronal glycoprotein, emerged as a molecule which promotes alpha synuclein internalization during the course of Parkinson’s disease. This uptake is N-linked glycans-dependent, which drew attention to the potential therapeutic value of αSacetyl-glycan interactions in the treatment of Parkinson’s disease. Neurexin 1b also takes part in the response to alpha-synuclein aggregation as shown in mouse primary neuronal culture [198].

In addition to the reactive and proactive regulatory changes, proteoglycan expression in Alzheimer’s disease is also regulated at the epigenetic level: a large-scale sex-specific meta-analysis of DNA methylation uncovered 14 proteoglycans, including tenascins, associated with the Braak stage in females, but not in males [199]. 

#### 4.2.4. Extracellular Matrix Modulating Enzymes and Neurodegeneration

A plethora of enzymes dynamically shapes the composition of brain extracellular matrix. They either are activated because of excess matrix depletion or emerge as the trigger of pathology in neurodegeneration. Whilst many studies report about the effect of MMPs, the exact source remains unexplored. Thus, neurons, astrocytes, oligodendrocytes, microglias as well as pericytes can equally produce MMPs, but the trigger, timing and select cellular source of release in the different neurodegenerative diseases remain ambiguous. Obviously, human studies can offer only limited insight into molecular mechanisms and report rather about altered MMP levels in disease. Further, most of the in vitro and animal studies focus on the triggers or effects of increased or decreased MMPs expression/release and on its monitorable and measurable pathohistological or pathobiochemical effects. Without going into detailed cell biological discussion about cell-specific molecular regulation of MMPs’ production and release in the healthy brain, we focus on and summarize the relevant information which is related to neurodegenerative processes. In addition to diagnostically useful data about MMPs expression in human clinical studies, we also summarized known molecular mechanisms explored in neurodegenerative disease models in vitro and in vivo.

Enzymes belonging to the TIMP family form non-covalent bonds with matrix metalloproteases to modulate their function [60]. TIMP1 and TIMP 2 emerge as useful diagnostic markers in neurodegenerative diseases: TIMP 1 level elevates in the CSF in Alzheimer’s and Parkinson’s diseases as well as in amyotrophic lateral sclerosis (ALS), while TIMP-2 levels increase in Alzheimer’s and Huntington’s diseases.

In multiple sclerosis, transient upregulation of extracellular matrix molecules reflects a physiological response. Whereas MMPs help in the remyelination processes, their dysregulation or dysfunction leads to insufficient extracellular matrix degradation/accumulation, with consequential demyelination and disruption of the blood–brain barrier. In this action, activation of astrocytes, microglia and macrophages are critical; additionally, they help inflammatory cell attraction and invasion. The complex action of the different MMPs are summarized in Table 1. 

Changes in the level of MMPs carry diagnostic value: elevated MMP-9 levels were detected in the CSF of patients with relapsing-remitting multiple sclerosis and patients suffering from the primary progressive form of the disease showed similar results [214]. MMP-9 concentration was also elevated in the serum of ALS patients. This change, however, is likely caused by peripheral nerve and muscle damage, as MMP-9 level in the CSF did not increase [215].

The role of tissue enzymatic activity in Alzheimer’s disease emerged from the basic idea that the amyloid precursor protein is processed by α-secretases of the ADAM family in physiological conditions, which show structural and functional similarities with MMPs. In human tissue, MMP-9 expression increased already in moderate-stage Alzheimer’s disease, whereas MMP-2 expression increased only in late-stage Alzheimer’s disease [216]. Other studies using immunohistochemistry and MMP activity assays on human frontal cortical brain samples reported no changes in the levels and activities of MMP-2, -3 and -9, which were unrelated to β-amyloid load [217]. A series of in vitro and rodent in vivo studies have shown that β-amyloid is a substrate of MMPs and can also trigger their expression [218,219,220]. Indeed, human studies typically report about increased MMP levels in Alzheimer’ disease, which—according to their types and triggered mechanisms—can have beneficial or detrimental effects on disease development [218]. In particular, MMP-9 attracted much attention, first when it was shown to cleave β-amyloid in vitro [221], followed by the clinical phenomenon that its plasma level is typically increased in Alzheimer’s disease patients [60,221]. Neuromorphologists soon unveiled that MMP-9 is also involved in perineuronal net maintenance, and that elevated MMP-9 levels may contribute to both perineuronal net degradation and altered levels of β-amyloid in Alzheimer’s disease [222,223].

The causal role, its reactive expression change and the diagnostic use of MMPs are still ambiguous. In mice, intracerebroventricular injection of several β-amyloid fragments increase MMP-9, but not MMP-2 activity and protein expression in hippocampal neurons and glias [224]. Because MMP inhibitors alleviate cognitive impairment in vivo as well as neurotoxicity in vitro, MMP-9 likely plays a causal role in β-amyloid-induced cognitive impairment and neurotoxicity [224]. Indeed, neuronal overexpression of MMP-9 in the 5xFAD transgenic mouse model increased sAPPα levels and decreased β-amyloid oligomers; however, amyloid plaque load in the brain was not affected [225]. In vitro, MMP-9 constitutes an endogenous islet protease that limits islet amyloid deposition and its toxic effects via degradation of the deposit [226]. Both recombinant human MMP-2 and MMP-9 generate β-amyloid fragments from Aβ40 and Aβ42; these fragments are soluble and neither exhibit fibrillogenic properties nor induce cytotoxicity in human cerebral microvascular endothelial or neuronal cells, which suggests their role in clearance, but not in fibrillogenesis [227]. Of note, human iPSCs from patients with sporadic Alzheimer’s disease (sAD) and familial Alzheimer’s disease (fAD) enhance MMP-2 and MMP-9 production, which leads to tau degradation, in both its monomeric and pathologically aggregated forms [228]. MMP-2 and MMP-3 [229] as well as MMP-7 [230] cleave soluble β-amyloid, and both MMP-9 and MT1-MMP [231] are able to degrade soluble and fibrillar forms of β-amyloid. Further, MMP-2 expression increases and co-localizes with tau pathology in the entorhinal cortex at early stages of AD-related pathology, which is likely a response geared to eliminating production of toxic truncated tau species in Alzheimer’s-diseased brains [229]. Tau is also a substrate of both MMP-3 and MMP-9, shaping tau aggregation behavior [232]. Experiments in transgenic mouse strains showed that MT5-MMP uses an alternative pathway: it promotes neuroinflammation and neuronal excitability, which leads to and β-amyloid production both in vitro [233] and in vivo, leading to cognitive decline [234]. MMP-3, in turn, triggers NGF dysmetabolism, leading to cholinergic atrophy and cognitive deficits in a sex-specific manner [235].

Although MMP-9 seems to play a casual role in disease development, its increased expression emerges as a reactive response. Cerebrovascular β-amyloid induces expression and activation of MMP-9 in cerebral vessels, and topical application of recombinant MMP-9 results in a time- and dose-dependent cerebral hemorrhage [236]. Further, experimental cerebral microemboli increase β-amyloid protein deposition and astrocytic MMP-9 production in APP/PS1 transgenic mice, which reflects imbalance between protein synthesis and removal. β-amyloid upregulates MMP-2 expression through the ERK and JNK signaling pathways in brain endothelial cells, which leads to enhanced vascular inflammatory stress and, therefore, cerebral amyloid angiopathy [237]. In astrocytes, the regulation of MMP-2 is complex: oligomeric β-amyloid downregulates MMP-2 expression but induces proinflammatory cytokine production, which stimulates MMP-2 release [238]. Nitric oxide, a critical signaling molecule in inflammation pathogenesis, increases MMP-9/TIMP-1 ratios to enhance the degradation of fibrillar β-amyloid in vitro and in vivo [239]. In an alternative inflammatory response, the β-amyloid fragments 25–35 trigger pro-MMP-9 release from human neutrophils of pro-MMP-9 [240]. Hyperoxygenation treatment upregulates MMP-2, MMP-9 and tPA expression which reduces β-amyloid accumulation and rescues cognitive impairment in APP/PS1 transgenic mice [241]. Female hormones emerged as effective triggers: estrogen activates MMP-2 and MMP-9 to increase beta amyloid degradation mediated through its ERα receptor subtype—at least in SH-SY5Y human neuroblastoma cells [242]-and Medroxyprogesterone Acetatea, a progestogen used in hormonal contraception suppresses amyloid-beta degradation in an MMP-9-dependent manner in vitro, and potentially compromises the clearance of β-amyloid in vivo [243]. Androgens, like dihydrotestosterone, enhance the expression of CD147, an extracellular matrix metalloproteinase inducer, which blocks MMP-2 production, thereby increasing β-amyloid level [244]. Amongst neurotransmitters, histamin can be of therapeutic potential, since it induces MMP-9 production via the glial H1-receptor, at least in vitro [245]. Of note, anti- β-amyloid immunotherapy significantly activates the MMP-2 and MMP-9 proteinase degradation systems, which, however, likely leads to increased microhemorrhage occurrence [246]. Natural marine and terrestrial compounds can modulate MMP-2 and MMP-9 activity, forecasting therapeutic potential [247]. Simvastatin, a hydrophilic statin that could cross the blood–brain barrier, reduced MMP-9 expression, which correlated with cognitive function improvement in a rat model [248]. Notably, whilst MMP inhibitors like simvastatin do not have direct antioxidant effects, they not only reduce cerebral amyloid angiopathy but also oxidative stress [249].

## 5. Diagnostic Use and Medical Screening

Diagnostically, the investigation of CSF and plasma samples attracted attention. CSF concentrations of MMP-9 decrease, concentrations of MMP-3 increase, whilst neither MMP-2 nor TIMPs show significant changes in Alzheimer’s disease patients [250]. Other studies showed significantly decreased MMP-2 and MMP-3 levels in the CSF in samples with significantly reduced β-amyloid levels, which suggest that they are directly linked to β-amyloid in the brain, and a dysfunction may influence the processing of β-amyloid [251]. MMP-3 levels correlate to the duration of Alzheimer’s disease and with CSF T-tau and P-tau levels in the elderly controls [252,253]. When investigating plasma samples, a decrease of MMP-2 and MMP-10 [254], but no change in MMP-9 expression level, was reported in Alzheimer’s disease patients [255]. Serum MMP-2 in acute phase emerges as a promising biomarker: its level correlates with the recurrence of cerebral amyloid angiopathy/intracerebral hemorrhage, which helps to evaluate the risk of cognitive impairment [256]. Medical screening of MMPs offers further potential. High plasma MMP-9 levels increase the conversion risk between mild cognitive impairment (MCI) patients with and without ApoE4. Cognitively healthy individuals with risk markers for future Alzheimer’s disease, i.e., critical CSF biomarker levels of T-tau, P-tau and Abeta42, or the presence of the APOE epsilon4 allele, have higher CSF MMP-3 and MMP-9 levels and higher CSF MMP-3/TIMP-1 ratios compared to the healthy individuals without risk markers [257]. In a Slovakian cohort study, *MMP2* rs243865 and *MMP3* rs3025058 promoter polymorphisms were shown to impact on the onset of Alzheimer’s disease [258]. On the other hand, the TIMP-2 rs7503726 AA genotype was inversely correlated with Alzheimer’s disease susceptibility, and the presence of minor alleles of rs7503726 (A allele) had protective effects against Alzheimer’s disease.

## 6. Comments on Human versus Animal Studies

Obviously, valid data to understand neurodegenerative diseases shall derive from human studies. Brain areas typically affected in neurodegenerative diseases show different/higher phylogenetic maturation in humans, which poses a difficult challenge to find usable animal or in vitro models to model human pathology. Especially cell biological and regulatory processes call for experimental paradigms which cannot be carried out in humans. Thus, due to species differences, and despite thorough experiment planning and critical data analysis, results gained from animal models can be incomplete or misleading. In Table 2, we wish to highlight some examples of incomplete, contradictory or ambiguous results obtained from human/animal (animal cell) experiments.

## 7. Conclusions

The mainstream of neurobiology studies in the field of neurodegenerative diseases concentrate upon intra- and intercellular alterations, which manifest in changes of the physiological and morphological character of the neuron. We believe that understanding the fine balance in brain extracellular matrix composition provides a major contribution to understanding these changes and offers valuable diagnostic and therapeutic tools for scientists and clinicians alike. The enzymatic machinery of brain extracellular matrix emerges as an especially promising druggable target, where enzyme manipulation can shape the destruction or protection of characteristic matrix assemblies. Their in vivo diagnostic imaging as well as their selective therapeutic access are the major challenges which scientists and physicians currently face.

## Figures and Tables

**Figure 1 ijms-23-11085-f001:**
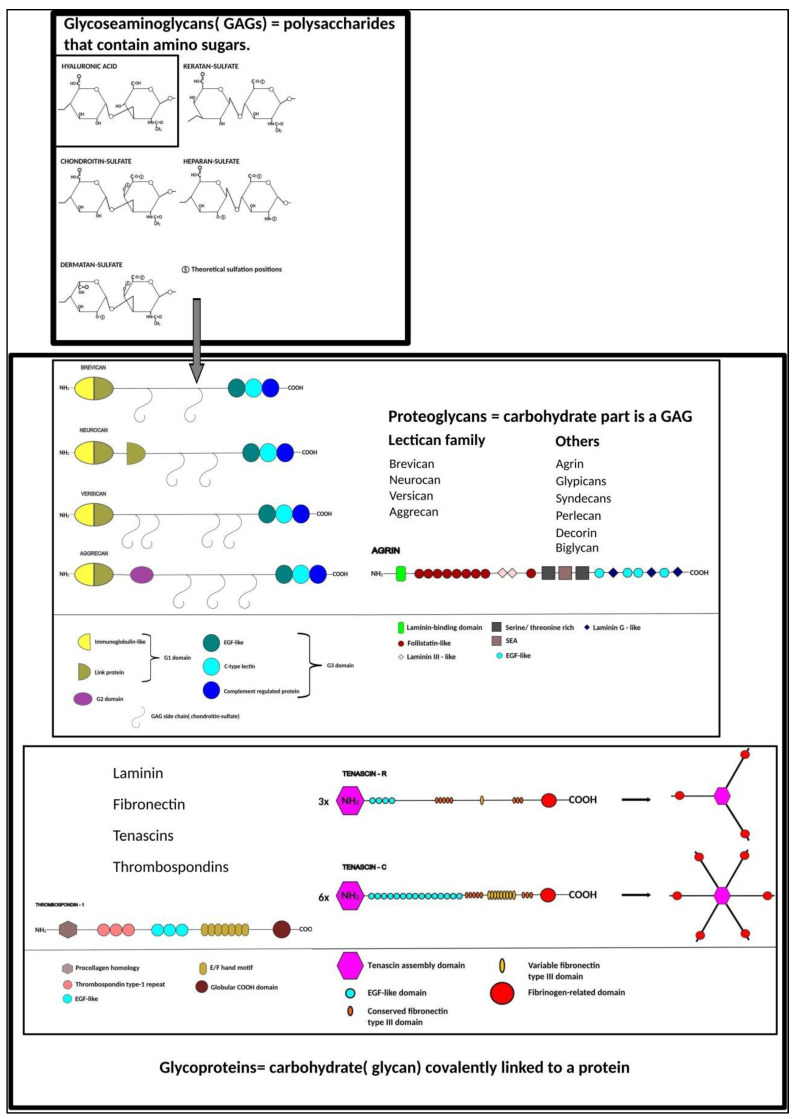
Typical components of central nervous system extracellular matrix “ground-substance”.

**Table 1 ijms-23-11085-t001:** Actions and characteristics of MMPs.

Name	Characteristics
MMP-3/stromelysin-1	- mainly produced by astrocytes [200] - generally upregulated following demyelination, exacerbates demyelination [201] - leads to disruption of blood–brain barrier [202] - helps remyelinization by CSPGs, fibronectin and myelin debris degeneration and increases the bioavailability of growth factors - activates other MMPs (e.g., MMP7, MMP9) [200,203]
MMP-12/macrophage elastase	- produced by microglia/macrophages during demyelination; [201] this activation persists in chronic lesions - produced by astrocytes during remyelination: cleaves extracellular matrix and stimulates remyelination [201] - cleaves osteopontin, which might induce cell death in activated T-cells: protective factor [204,205] - multiple sclerosis patients have lower levels of MMP-12 in their cerebrospinal fluid compared to control [206]
MMP-9/gelatinase B	- unaltered levels during demyelination [201], but probably takes part - upregulated in remyelination: oligodendrocyte process growth, aids oligodendrocyte maturation [201] - product of microglia, macrophages [207] - disrupts blood–brain barrier, helps inflammatory cells to enter the central nervous system [208]
MMP-2/gelatinase A	- constitutively expressed in the central nervous system [209,210] - probably takes part in blood–brain barrier disruption and demyelination [208]
MMP-7/matrilysin	- constitutively expressed in the central nervous system [208] - potential role: regulator of extracellular matrix turnover under physiological conditions [208] - role in multiple sclerosis is not fully understood; probably takes part in inflammatory cell extravasation, and might play a role in demyelination and axon loss [211,212,213]

**Table 2 ijms-23-11085-t002:** Role of brain extracellular matrix molecules—results gained from human versus animal studies.

Regarding	Animal Study		Human Study
	Species	Result	
Molecular weight dependent actions of HA in AD	Rat	Active low molecular weight heparin might be protective against AD pathology or even reverse amyloidosis [259].	High molecular weight heparin promotes the conversion of random coils to beta-sheets [175].
Contribution of HSPGs to AD pathology	Mouse neuronal cells, Chinese hamster ovary cells	HSPGs contribute to the cellular uptake of amyloid-beta, which is a clearance mechanism [260].	Agrin (a type of HSPG) is accumulated in an insoluble form in AD, likely taking part in amyloid-beta formation. Agrin might also have a role in microvasculature changes occurring in AD [261].
Level of protection offered by aggrecan-based PNNs against AD pathology	Mouse	Aggrecan effects tau protein synthesis and phosphorylation but does not protect against tau pathology per se [262].	Aggrecan-based PNNs protect cells from tau pathology [113].
Role of tenascins in acute and chronic neuroinflammation	Mouse	Tenascin-C contributes to the inflammatory aspect of AD and its functional inhibition lessens AD symptoms [189]. Tenascin-R restricts distribution and internalization of tau as a component of PNN [112].	Tenascin R and C were downregulated in acute MS plaques. Subacute and chronic plaques showed near-normal levels [186]. Tenascin-C is associated with cored plaques [190].

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
