# Peer review of "The Role of Extracellular Matrix in Human Neurodegenerative Diseases"

_ijms, 2022, doi:10.3390/ijms231911085_

Round 1

Reviewer 1 Report

The manuscript reviews existing literature on the roles of the extracellular matrix in neurodegenerative diseases. Authors analyze the evidence to find answers to some fundamental questions on which literature is largely ambiguous. The authors also explore the possibility of the clinical relevance of ECM and PNN-related studies. The overall goals and structure of the review manuscript are relevant and important. The majority of the sections are well written. However, from a reader’s point of view, a few things need to be addressed.

I have the following comments:

1.       The most apparent and important shortcoming is a critical analysis of the studies that were discussed. Most of the sections read like a description of studies without a critical analysis to provide a conclusive update to the readers.  Therefore, a conclusive statement/perspective of authors on the discussed literature in each section will be a great addition.

2.        In several places, I intended to go into details using the reference, however, either it was not present or was a review article itself. One of the examples: Page 4, lines 128-129, endogenous Chondroitinase is interesting but had no reference to learn more about it. Similarly, on page 7, lines 299-304 have interesting information on hyaluronan which readers might want to explore more by going through the original reference.     

3.       Without chemical structure and structural representations, figure 1 is no more than a bunch of words. I recommend adding images of chemical structures and organizations of GAGs.

4.       A table of ECM/PNN components and their ambiguous fate in neurodegenerative diseases with references and information on Human/animal models will be a great addition to this review.

5.       Since the review seems to have a clinical perspective, it would be important to explicitly distinguish between human and mouse model studies and discuss if any disagreements are present in the literature.

6.       MMPs have been discussed extensively however very little is discussed on their cellular source and their regulation. 

Author Response

Answers for Reviewer #1

Dear Reviewer,

thank you for critically commenting on our manuscript. Please find our point-to-point answers below. We hope that the present version fulfils your requirements.

  1. The most apparent and important shortcoming is a critical analysis of the studies that were discussed. Most of the sections read like a description of studies without a critical analysis to provide a conclusive update to the readers.  Therefore, a conclusive statement/perspective of authors on the discussed literature in each section will be a great addition.

Thank you for drawing our attention to this point. We improved our manuscript by including such paragraphs according to your wish. Thus, sections 3 and 4.1. end with a summary, whilst sections 4.2.2. and 4.2.4 begin with an extra paragraph. These parts highlight contradictions, missing data and suggest future research directions to complete our knowledge about brain extracellular matrix in select aspects.

  1. In several places, I intended to go into details using the reference, however, either it was not present or was a review article itself. One of the examples: Page 4, lines 128-129, endogenous Chondroitinase is interesting but had no reference to learn more about it. Similarly, on page 7, lines 299-304 have interesting information on hyaluronan which readers might want to explore more by going through the original reference.

We updated the reference list and original articles are now included in case they were missing in the original version of the manuscript. As a consequence, the reference list grew longer: we included over 40 further references.

  1. Without chemical structure and structural representations, figure 1 is no more than a bunch of words. I recommend adding images of chemical structures and organizations of GAGs.

We completed the table accordingly.

  1. A table of ECM/PNN components and their ambiguous fate in neurodegenerative diseases with references and information on Human/animal models will be a great addition to this review.

We have included a table (Table 2) with major typical examples of such contradictions/ambiguities between results gained from human and animal analyses/experiments.

  1. Since the review seems to have a clinical perspective, it would be important to explicitly distinguish between human and mouse model studies and discuss if any disagreements are present in the literature.

Please also consider our answer to point 4 here. Also, we went through the text to mention where results derive from (please note that all changes in the text are indicated yellow, including these).

  1. MMPs have been discussed extensively however very little is discussed on their cellular source and their regulation. 

We have included a paragraph at the beginning of chapter 4.2.4. to shortly address this issue. We did not have the intention to give a detailed description about the cellular regulation of MMP production and release. Instead, we focussed on those data which are related to MMPs levels and effects in disease. Please accept that many results/studies we cite are human studies, where cellular source and especially regulation process at the molecular level are difficult or not possible. Please note though, that cellular source as well as the biological trigger of their production is mentioned explicitly in Table 1 for several MMP types, and the text also includes information in this context (189-192), and molecular regulation is also discussed 193-196, with further identifying and discussing biological triggers (194, 196, 197, 198, 200, 201, 203). These latter data derive from in vitro or animal studies.

Sincerely yours,

Alán Alpár

Reviewer 2 Report

This review appears complete and exhaustive, assembling the essential discoveries about the relationship between the extracellular matrix and the development of neurodegenerative diseases. It deserves to be published.

Author Response

Dear Reviewer,

thank you for commenting on our manuscript. We are happy that you find our work worthy to be published.

Sincerely yours,

Alán Alpár